# Reductive Amination for LC–MS Signal Enhancement and Confirmation of the Presence of Caribbean Ciguatoxin-1 in Fish

**DOI:** 10.3390/toxins14060399

**Published:** 2022-06-09

**Authors:** Fedor Kryuchkov, Alison Robertson, Elizabeth M. Mudge, Christopher O. Miles, Soetkien Van Gothem, Silvio Uhlig

**Affiliations:** 1Toxinology Research Group, Norwegian Veterinary Institute, P.O. Box 64, 1431 Ås, Norway; fedor.kryuchkov@vetinst.no; 2School of Marine and Environmental Sciences, 600 Clinic Drive, Mobile, AL 36688, USA; arobertson@disl.org; 3Marine Ecotoxicology, Dauphin Island Sea Laboratory, 101 Bienville Blvd., Dauphin Island, AL 36528, USA; 4Biotoxin Metrology, National Research Council, 1411 Oxford Street, Halifax, NS B3H 3Z1, Canada; elizabeth.mudge@nrc-cnrc.gc.ca (E.M.M.); christopher.miles@nrc-cnrc.gc.ca (C.O.M.); 5Department of Bioanalysis, Faculty of Pharmaceutical Sciences, Ghent University, Ottergemsesteenweg 460, 9000 Gent, Belgium; soetkien@gmail.com

**Keywords:** ciguatoxin, Caribbean ciguatoxin, Girard’s reagent, quaternary ammonium, LC–MS, derivatization

## Abstract

Ciguatera poisoning is a global health concern caused by the consumption of seafood containing ciguatoxins (CTXs). Detection of CTXs poses significant analytical challenges due to their low abundance even in highly toxic fish, the diverse and in-part unclarified structures of many CTX congeners, and the lack of reference standards. Selective detection of CTXs requires methods such as liquid chromatography coupled to tandem mass spectrometry (LC–MS/MS) or high-resolution MS (LC–HRMS). While HRMS data can provide greatly improved resolution, it is typically less sensitive than targeted LC–MS/MS and does not reliably comply with the FDA guidance level of 0.1 µg/kg CTXs in fish tissue that was established for Caribbean CTX-1 (C-CTX-1). In this study, we provide a new chemical derivatization approach employing a fast and simple one-pot derivatization with Girard’s reagent T (GRT) that tags the C-56-ketone intermediate of the two equilibrating C-56 epimers of C-CTX-1 with a quaternary ammonium moiety. This derivatization improved the LC–MS/MS and LC–HRMS responses to C-CTX-1 by approximately 40- and 17-fold on average, respectively. These improvements in sensitivity to the GRT-derivative of C-CTX-1 are attributable to: the improved ionization efficiency caused by insertion of a quaternary ammonium ion; the absence of adduct-ions and water-loss peaks for the GRT derivative in the mass spectrometer, and; the prevention of on-column epimerization (at C-56 of C-CTX-1) by GRT derivatization, leading to much better chromatographic peak shapes. This C-CTX-1–GRT derivatization strategy mitigates many of the shortcomings of current LC–MS analyses for C-CTX-1 by improving instrument sensitivity, while at the same time adding selectivity due to the reactivity of GRT with ketones and aldehydes.

## 1. Introduction

Ciguatera poisoning is caused by the ingestion of seafood contaminated with algal-derived ciguatoxins (CTXs) (Figure 1). It is a global problem in circumtropical regions and can cause health impacts in temperate regions through export of contaminated fish. Epi-benthic dinoflagellate algae of the genera *Gambierdiscus* and *Fukuyoa* [1] have been identified as the source of CTXs in the Pacific Ocean [2,3,4,5,6]. The algal toxins, or presumed CTX precursors, bioaccumulate in marine biota and are thought to undergo metabolic biotransformations as they move between species and trophic levels in reef food webs. Historically, CTXs have been categorized into three groups (Pacific, Caribbean, and Indian Ocean) based on their first reported geographical origin, though this classification scheme is somewhat outdated and incongruent with naming conventions of other marine toxin classes [7]. The most studied CTXs are the Pacific CTXs (P-CTXs) that are further divided into two subtypes (type I and type II) [8,9], and are currently represented by 22 analogs [10]. The least studied group are the Indian CTXs (I-CTXs), which so far include five analogs that were partially characterized by LC–HRMS methods [11,12,13]. Caribbean CTXs (C-CTXs) were initially discovered in fish from the Caribbean Sea [14] and to date 12 C-CTXs have been reported [15], but only four have been structurally characterized [16,17]. All elucidated ciguatoxins (CTXs) have been shown to contain a ladder-shaped polyether structure and a relatively high molecular mass (1022–1158 Da) [8,9,10,18,19]. Ciguatoxic fish in the Western Atlantic Ocean, including the Caribbean and Gulf of Mexico, and more recently in southern Europe [20,21,22], have been reported to be predominantly contaminated with C-CTX-1 and -2, which is an equilibrating pair of 56-epimers (**1**, Figure 1, based on [18]).

Several in vitro bioassays are currently in use for the detection of CTXs, typically in combination with targeted low-resolution LC–MS/MS methods. The most commonly reported bioassay for CTX assessment is the sodium-channel-dependent 3-(4,5-dimethylthiazol-2-yl)-2,5-diphenyl-2H-tetrazolium bromide (MTT)-based mouse neuroblastoma (MTT-N2a) assay. The MTT-N2a method is a tetrazolium-based assay that provides a measure of mitochondrial metabolic activity in N2a cells exposed to sample extracts, which indirectly correlates to cell viability [23,24,25]. Some selectivity is afforded by the co-exposure of cells to ouabain and veratridine (or veratrine), which allows differentiation of toxins acting on voltage-gated sodium channels (Na_V_) e.g., CTXs, brevetoxins etc., from phycotoxins with alternative toxicity mechanisms (e.g., okadaic acid, domoic acid). Modifications to the MTT-N2a assay can also allow functional detection of toxins that inhibit Na_V_ such as saxitoxins and tetrodotoxin [26,27], and toxins that exert their toxicity through Ca^2+^ modulation such as maitotoxin [28]. This high-throughput assay is extremely sensitive and can be applied to the evaluation of CTX activity in both algal and fish extracts in ciguatera investigations [29,30,31,32,33,34]. The MTT-N2a assay enables the detection of CTXs at or below their respective US FDA-recommended guidance levels (0.01 µg/kg CTX-1B [35], and 0.1 µg/kg C-CTX-1, in fish [36,37], which are also endorsed by the European Food Safety Authority [38]). Other reported approaches include radioligand [29,39] and fluorescence-based receptor binding assays [29,40,41,42], and immunoassays [10,43], that have been reviewed by others [10,44,45].

While most of the in vitro bioassays for CTX detection are very sensitive and can estimate the risk of fish by measuring a composite “ciguatoxicity”, they lack the ability to discriminate between CTX congeners. For such analyses, liquid chromatography coupled with low-resolution tandem mass spectrometry (LC–MS/MS) or high-resolution mass spectrometry (LC–HRMS) are the methods of choice. Currently, only methods based on modern LC–MS/MS instruments, such as triple quadrupole [46,47,48] or quadrupole ion-trap technologies [49], can detect levels below the FDA guidance level (limit of quantitation (LOQ) in the range 0.01–0.1 µg/kg for P-CTXs and 0.01 µg/kg for **1**). LC–HRMS instruments are able to perform one or several types of MS/MS experiments and provide information regarding accurate masses, isotopic distributions, and adduct ions, and also allow retrospective analyses of full-scan data. LC–HRMS instruments, such as the quadrupole time of flight (Q-TOF) [50] or Orbitrap [51] instruments, are generally more expensive, and cannot compete with low-resolution MS/MS instruments in terms of detection limits, which are one–two orders of magnitude higher than is the case for state-of-the-art LC–MS/MS.

Here we report that derivatization of **1** with Girard’s reagent T (GRT), a positively-charged quaternary ammonium tag, improves the analytical sensitivity of LC–MS detection in fish extracts by more than an order of magnitude (Figure 2).

## 2. Results and Discussion

CTXs are analytes with poor ionization efficiency [46,52]. Recent publications showed limits of quantification for P-CTXs in the range 1–10 pg on-column using LC–MS/MS, depending on the instrument used [46,52]. There are three features that impair detection of **1** at low concentrations by mass spectrometry: the MS signal intensity is spread over a complicated set of adduct ions and ions from dehydration (Figure 3V); the absence of functional groups of sufficient acidity or basicity to promote ionization, and; **1** is a pair of equilibrating C-56 epimers (Figure 1) that elute as a rather broad peak (Figure 3II) from LC columns [17]. Although the complex mass spectra of C-CTXs are useful for identification of novel congeners, they represent an obstacle for instrument sensitivity and quantification since the relative abundance of adducts can change over time and is dependent on the instrument status. HRMS-based software may sum peak areas for all detected adducts and report a “total peak area”. However, such an approach is not feasible for low-resolution instruments because these rely on selected MS/MS transitions specifically defined for each ion of interest. Tagging an analyte molecule with a constant charge is common practice in mass spectrometry, and numerous methods have been published for the improvement of ionization efficiency of oligosaccharides [53], peptides [54], steroids, [55] and lipids [56].

We have recently studied and described the epimerization of **1**, which can lead to asymmetrical and rather broad LC peaks, especially when octadecylsilane (ODS)-based column chemistries are employed [17]. We therefore studied the derivatization of the equilibrating hemiketal/ketone group at C-56 of **1** (Figure 1) via reductive amination with a hydrazide (GRT) and an amine ((2-aminoethyl)trimethylammonium chloride hydrochloride, AETMA), each of which contains a quaternary trimethylammonium ion (Figure 2). The aim of this approach was to insert a fixed charge into the analyte while simultaneously removing the possibility for ketone–hemiketal equilibration. Reducing reagents were selected that were expected to be able to preferentially reduce the C=N bond of imines and hydrazones in the presence of a keto-group. Reduction of hydrazone **3** to yield **4** (Figure 2, Table 1) was found to be necessary, as **3** was unstable on-column resulting in poor chromatography, likely in part because of hydrolysis. These approaches were modifications of derivatization techniques that have been reported for the LC–MS based quantification of oligosaccharides, which are well known for their low LC–MS ionization efficiencies [53].
Figure 2Reaction scheme showing conversion of C-CTX-1 (**1**) to the desired GRT hydrazine **4** via GRT hydrazone **3**, and possible routes for formation of by-products **2** and **5**. The structures of GRT and AETMA are shown in the inset.
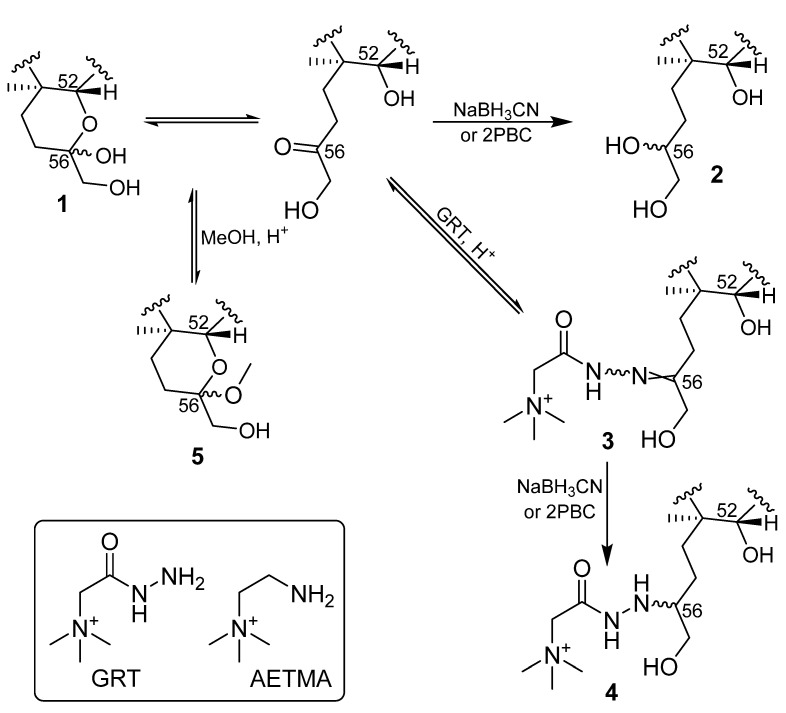


### 2.1. Optimization of the Reductive Amination of ***1***

Achievement of efficient derivatization requires thorough optimization of reaction conditions. Parameters such as pH, concentration of acid/base, and reaction temperature and time, as well as whether the reductive amination is performed as a one-step or two-step procedure, are important for the derivatization yield (Appendix A) [53]. Due to the very limited availability of highly ciguatoxic materials containing **1** in the concentrations required for reliable LC–HRMS detection, the number of replicate measurements during method development had to be kept to a minimum.

We found that **1** could be derivatized by AETMA catalyzed by acetic acid (derivatization method 3), as has been reported for oligosaccharides [57,58], but with only modest efficiency (typically <50%). However, although derivatization with AETMA produced a derivative with a fixed positive charge (Figure 2), the product was a pair of epimers in a ca. 1:1 ratio (Appendix A). This results in reduced signal/noise, especially for low-level samples. The results obtained with AETMA demonstrated in principle the potential of this derivatization approach, so we examined derivatization with GRT to determine whether it could be a more suitable reagent for analysis of **1**.

Initial experiments, prior to optimization, used cyanoborohydride as the reducing agent, which was added to a mixture of ciguatoxic extracts, GRT, and 10% formic acid. After shaking the mixture for 2 h at 60 °C, we were able to detect five CTX peaks in the LC–HRMS chromatograms (Appendix A). These could be attributed to unreacted and chromatographically unresolved C-CTX-1 epimers (**1**), the non-reduced GRT-hydrazone of **1** (**3**), and the desired C-CTX-1–GRT reduction products (**4**), as well as C-CTX-3/-4 (**2**) as undesired by-products from reduction of **1**. We also detected a compound affording *m*/*z* 1123.6231 ions without the presence of other prominent adducts in the reaction mixture (Appendix A). This ion originated from the recently reported artefact C-CTX-1 56-methylketal (**5**) (Figure 2 and Appendix A) [59], and was a result of in-source fragmentation (Appendix A). Confirmation of this was obtained by dissolving small amounts of dried sample-L in either methanol, 2-methoxyethanol, or 1-butanol, with the addition of 10% trifluoroacetic acid (TFA) (Appendix A). The products from all three reactions were observed at *m*/*z* 1123.6231 (as the predominant ion) but at different retention times. These retention times varied with the alcohol that had been used as the solvent, with retention times for methanol < 2-methoxyethanol < 1-butanol (Appendix A), indicating the formation of the 56-methyl, -(2-methoxyethyl), and –butyl ketal derivatives of **1**. Furthermore, compound **5** eluted as a sharp peak from the UHPLC column, and thus the two expected 56-epimers were either not chromatographically resolved or the formation of one epimer is preferred during the reaction with alcohol. Another observation was that **5** did not form sodium or ammonium adducts in positive mode (Appendix A) and was not detectable in negative mode (data not shown). This behavior is attributable to the absence of the vicinal diol moiety. Another peak in the chromatograms was due to the non-reduced hydrazone intermediate **3** (Figure 2). Increasing the reaction temperature and replacing formic acid with TFA led to faster conversion, but lower overall reaction yield (Appendix A). As a compromise between a long reaction time (i.e., up to 22 h) and pronounced formation of **5**, we found optimum conditions for reductive amination of **1** with GRT using cyanoborohydride as the reducing agent in a 22-h two-step reaction (derivatization method 2).

According to data on the reductive amination of oligosaccharides [57], using 2-picoline-borane complex (2PBC) for reduction of the hydrazones provides higher yields and shorter reaction times. Replacing cyanoborohydride with 2PBC for reductive amination of **1** with GRT eliminated the formation of side-product **5**, most likely due to faster reduction of intermediate **3**, leading to a shift in the equilibrium position and preferentially to formation of the stable and thermodynamically favored product **4** (Figure 2). This allowed use of a stronger acid (TFA) and an increase in the reaction temperature to 60 °C, which led to a faster one-pot derivatization (derivatization method 1A). TFA and formic acid were found to be about equally efficient in catalyzing the derivatization reaction, but they differ in the formation of side products. We observed that the TFA-catalyzed reaction resulted in a slightly lower production of side-products **2** and **5** compared to reaction employing formic acid (Appendix A). However, we could not see any substantial difference in the signal enhancement between the two acids (i.e., derivatization methods 1A and 1B). A potentially problematic property of TFA is its ability to form ion pairs, which we observed when the TFA from the sample co-eluted with **4** or side-products such as **2**. This may lead to disturbance of the chromatography under some conditions, especially when faster gradients with short retention times are used, leading to the appearance of artefact peaks and poor peak shape (Appendix A). When TFA is used (derivatization method 1A), we recommend using longer gradients (with a retention time for **4** of at least 7 min). Alternatively, preparation of samples according to derivatization method 1B (i.e., using formic acid instead of TFA) can be used (Appendix A).

The very poor peak shape observed for **1** during LC–HRMS on ODS-based LC columns has been attributed to on-column epimerization of the 56-ketal [17]. The possibility for this epimerization is eliminated by reductive amination of **1** with GRT, and we observed excellent chromatographic peak shape and separation of **4** using an ODS column when using LC–HRMS method 2 (Appendix A). Thus, the derivatization procedure described here also opens the way for the application of a wider range of LC columns to the analysis of samples containing **1**.

### 2.2. Signal Enhancement of ***1*** Using GRT

Twelve fish extracts containing C-CTXs (A–L), and a negative control sample (N), were selected to test the procedure for derivatization and enhanced detection of **1**. Table 2 summarizes details of the signal enhancement, including peak areas for **1** and **2** prior to derivatization (from direct analysis), and peak areas for **2** and **4** after derivatization. The yield of the derivatization reaction and formation of by-products was studied in more detail for the samples containing the highest concentrations of **1** (A–D and L) by MS. Under optimized conditions (derivatization method 1A), we did not observe the presence of **5**, nor of unreduced intermediate **3**. The only by-product was **2**, which was also present naturally in the toxic extracts. On average, the ratio between major and minor epimers of **4** was in range 15:1–25:1 (Figure 3). In contrast, the epimer ratio of AETMA-tagged **1** was approximately 1:1 (Appendix A). To test the effect of the derivatization on instrument sensitivity, we diluted a CTX-containing fish extract (sample L) with variable proportions of an extract that did not contain detectable amounts of **1** (sample N) (Figure 4). Without derivatization, we were able to detect **1** in a mixture comprised of 20% sample-L in sample-N using LC–HRMS method 1, whereas after GRT derivatization we could reliably detect **1** (as derivative **4**) in a mixture comprised of 1% sample-L in sample-N. The signal enhancements observed in the dilution study (Figure 4) suggest that derivatization of **1** with GRT resulted in an approximately 17-fold increase in analyte peak area, consistent with the enhancements observed for the five selected extracts (samples A–D, and L) in Table 2. However, Table 2 also shows that the apparent signal enhancement for the different samples was somewhat variable (peak area increase by a factor of ~8–23). Thus, for routine applications, the achieved signal enhancement should be validated and refined. The observed variation of the signal enhancement for **1** via derivatization with GRT can in part be explained by very low peak areas for **1** in some cases, as well as variable matrix effects between samples by MS analysis.
Figure 3LC–HRMS (method 1) chromatograms and spectra of C-CTX-containing sample D: (**I**), Total ion chromatogram (TIC, *m*/*z* 1050–1350); (**II**), extracted ion chromatogram ([M+H–H_2_O]^+^ of **1**, *m*/*z* 1123.6200); (**III**), TIC (*m*/*z* 1050–1350) after derivatization with GRT and reduction (derivatization method 1), note that **2** is not visible as a result of the increased intensity of **4** vs. **1**; (**IV**), extracted ion chromatogram ([M]^+^ of **4**, *m*/*z* 1256.7415); (**V**), LC–HRMS spectrum of **1**; (**VI**), LC–HRMS spectrum of **4**. Note the ratio of epimers of **4** in panels (**III**,**IV**).
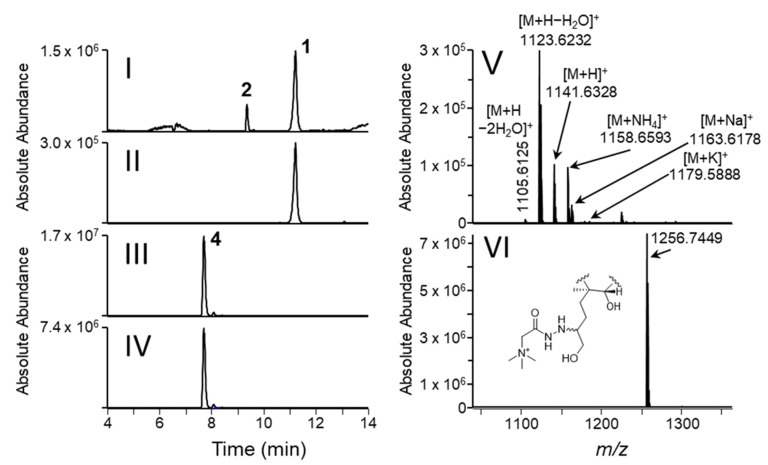


### 2.3. Application of Reductive Amination of ***1*** for LC–MS/MS Based Analysis

Product ion spectra of **4** were evaluated both using HRMS/MS (Appendix A), and on a triple quadrupole mass spectrometer to establish the applicability of this derivatization on low resolution instrumentation. The product ion observed was the quaternary trimethylammonium ion at *m*/*z* 60.4 which retained the positive charge (Figure 5). No other product ions were observed across the mass range and collision energy limits of the instrument; therefore this ion was selected for the MRM transition of **4**. LC–MS/MS source- and compound-specific conditions were optimized to ensure high sensitivity of **4** based on this transition.

Chromatographic comparison of the underivatized sample D, diluted to be consistent with the GRT-derivatized solution, revealed a significant enhancement in instrument sensitivity. The peak area response was 40 times higher for **4** compared to **1** in the underivatized sample (Appendix A). GRT-derivatized sample D was serially diluted with MeOH, and **4** was detected in all but the lowest dilution. The 1/1000-fold dilution gave a S/N of 80, which is well above recommendations for quantitation, while the peak detected in the 1/10 000-fold dilution had a S/N of 5, suggesting that it is above the limit of detection, but below quantitation levels. Linear response was confirmed with the serially diluted samples, with a linear range over several orders of magnitude from the non-diluted sample down to the 1/1000-fold diluted sample (Figure 6), and repeatability of replicate injections (*n* = 3) had a relative standard deviation of 0.5% in sample D. Given the lack of reference standards of C-CTXs, it is not possible to definitively assign a linear range or quantitative detection limit, but this data suggests the applicability of this derivatization technique to detect the presence of low levels of C-CTXs using low resolution mass spectrometers.

## 3. Conclusions

Reductive amination is a promising strategy for improving the signal response for C-CTX-1 in mass spectrometry-based detection. Adding a fixed charge to C-CTX-1 via derivatization with GRT increased instrument response by more than an order of magnitude. For routine applications, the derivatization reaction must be validated, especially with regard to matrix effects for determination of **1** and **4**, which will influence the apparent signal enhancement across different samples. Such a validation requires access to analytical standards and/or characterized reference materials. GRT was superior to AETMA, since the pair of epimeric GRT-derivatized products showed a large predominance of one epimer, which is an advantage for signal enhancement. Moreover, derivatization with GRT in MeOH–TFA did not result in the detectable formation of the methyl ketal by-product **5** under optimized conditions. Derivatization of **1** with GRT eliminated the possibility of on-column epimerization of the N-ring hemiketal moiety observed for **1**, and consequently the derivatization products were resolved as sharp symmetrical peaks not only on a pentafluorophenyl-propyl column, but also on an ODS LC column. Application of the derivatization reaction allowed reliable detection of **1** by LC–HRMS in fish tissue, and the reaction of **1** to form **4** provides additional confirmatory evidence of the presence of C-CTX-1 in complex matrices.

## 4. Materials and Methods

### 4.1. Chemicals and Reagents

Methanol (gradient quality) was from Romil Ltd. (Cambridge, UK). Acetone, water, hexane, chloroform, and dichloromethane used in extractions were all HPLC grade from Fisher Scientific (Hampton, NH, USA). Acetonitrile and water for instrumental analyses were Optima LC–MS grade (Fisher Scientific). Trifluoroacetic acid (TFA) (≥99.0%), formic acid (for LC–MS LiChropur, 97.5–98.5%), 2-methylpyridine borane complex (2PBC) (95%), sodium cyanoborohydride (reagent grade, 95%), 2-methoxyethanol (ReagentPlus, ≥99.0%), 1-butanol (ACS reagent, ≥99.4%), Girard′s reagent T (GRT, (hydrazinocarbonylmethyl)trimethylammonium chloride, LiChropur, 99.0–101.0%), and (2-aminoethyl)trimethylammonium chloride hydrochloride (AETMA) (99%) were from Sigma–Aldrich (St. Louis, MO, USA).

### 4.2. Sample Collection, Preparation, and Extraction

A variety of high trophic level fish including independent specimens of *Scomberomorus cavalla*, *Scomberomorus regalis*, and *Sphyraena barracuda* were collected on hook and line from St. Thomas, U.S., Virgin Islands, Puerto Rico, and Dauphin Island, Alabama, USA from 2016–2020 and used in this study (Table 3). All fish species were identified based on known morphological characteristics upon collection and stored on ice until return to the laboratory. Fish were then dissected within 4 h, or frozen whole until dissection could be conducted in a controlled laboratory environment. Muscle tissue was obtained from each side of the fish and all bones, skin, scales, and connective tissue removed. Excised muscle tissue was gently rinsed in deionized water and blotted dry with paper towels before homogenizing in a heavy duty air-cooled stainless steel meat grinder (STX-4000-TB2-PD fitted with a small grinding plate with 88 × 4 mm diameter holes; STX International, Lincoln, NE, USA). Individual samples were passed through the grinder at least three times, mixing tissue well between processing steps. Homogenized tissue was subsampled and stored at −20 °C until toxin extraction. All components of the grinder were soaked in warm water, then washed three times (detergent in water, 10% bleach solution, and water), and sterilized thoroughly between samples with 70% (*v*/*v*) isopropanol in water. Process controls (verified negative fish) were passed through the system periodically to evaluate crossover, which was not detected. To ensure consistency in sample identification from field identification and subsequent partitioning and subsampling of tissues, all samples were further identified by DNA barcoding methods [60] immediately prior to use. Reported species identifications for all samples used in this study (Table 3) included positive identification by morphometric and DNA barcoding methods.

Samples were extracted using one of three methods (Table 2). *Extraction 1*: Approximately 50 g of homogenized tissue was weighed and extracted in cold acetone (5 mL/g; 250 mL) in a stainless steel explosion-proof blender (Waring, McConnellsburg, PA, USA) at 8000 rpm, then quantitatively transferred and filtered through Whatman filter papers (20–25 µm, #4, and; 2.5 µm, #5, stacked on top of each other) in a Büchner funnel under light vacuum. The residual tissue recovered from the filters was then extracted an additional two times with 250 mL acetone, and filtrates from individual fish were pooled separately and stored at −20 °C in a sealed glass container overnight. Samples were subsequently re-filtered (Whatman #5; 2.5 µm) to remove protein and lipid precipitates, and rotary evaporated (Heidolph; Wood Dale, IL, USA) at 45 °C. The dried residues were then dissolved in 200 mL 80% aqueous MeOH and partitioned twice with an equal volume of hexane (200 mL). The methanolic phase was then adjusted to 50% by addition of 120 mL water (320 mL total extract volume), partitioned with chloroform (3 × 0.5 vol.; 160 mL), and the chloroform phase was rotary evaporated. Dried residues were dissolved in dichloromethane (5 mL) to transfer from the round bottom flasks and quantitatively transferred to glass tubes, evaporated under a stream of nitrogen at 35 °C, and stored at −20 °C. Final dried residues were diluted in 2 mL dichloromethane prior to SPE. Excessive frothing of the extract was observed during the initial phase of rotary evaporation (presumably caused by increased lipid extraction by acetone compared to MeOH), so a modified procedure (Extraction 2) was developed.

*Extraction 2*: Fish tissue homogenates (approx. 50 g) were weighed and extracted in MeOH (5 mL/g; 250 mL total), yielding an extract with approximately 15% water in the crude extract derived from the fish tissue (i.e., approx. 37.5 mL water from a 50 g sample). The wet tissue derived water was estimated from mean weights pre- and post-freeze-drying in parallel experiments with those fish species that consistently showed a 75% water loss (data not shown). Tissue was separated by filtration as described for *Extraction 1* and extracted a second time with 80% aq. MeOH at the original tissue to solvent ratio (250 mL). Filtrates from both extractions were pooled (500 mL at ~82.5%) for each fish sample and the volume adjusted with 15 mL water to bring the MeOH content of the extract to approx. 80% (515 mL). The extracts were immediately partitioned with hexane (2 × 0.6 vol.; 309 mL) and the hexane discarded. The water content of the methanolic phase was adjusted with water to 50% (309 mL added, total volume 824 mL), and the extracts were further partitioned with dichloromethane (3 × 0.5 vol.; 412 mL). The dichloromethane phases were collected, pooled, and rotary evaporated at 35 °C. Dried residues were dissolved in dichloromethane (5 mL), quantitatively transferred to glass tubes, evaporated under a stream of nitrogen at 35 °C, and stored at −20 °C. Prior to SPE, residues were diluted in 2 mL dichloromethane. Extraction method 2 expedited the partitioning and evaporation steps).

*Extraction 3*: Extraction was performed using freeze-dried and powdered *S. barracuda* (sample N) that had been verified by MTT-N2a and LC–MS/MS as not possessing ciguatoxic activity or containing **1**, respectively (data not shown). An aliquot (5 g dry weight; note: equivalent to 20 g wet weight as measured pre- and post- freeze dry) was suspended in 10 mL acetone–water (3:1, *v*/*v*) and gently shaken on an orbital shaker for 20 h at ambient temperature (~22 °C). The mixture was centrifuged at 15,000× *g* for 5 min, and the extract was evaporated to dryness under a stream of nitrogen at 60 °C. The residue was dissolved in 1 mL MeOH, partitioned with hexane (3 × 1 mL), then evaporated to dryness (60 °C, nitrogen), and finally diluted in 2 mL dichloromethane prior to SPE.

*SPE fractionation*: The dichloromethane solutions were loaded by gravity onto Silica SPE cartridges (Agilent Bond Elut, 1 g, 6 mL, 40 µm; Santa Clara, CA, USA), preconditioned with three column volumes (18 mL) each of MeOH and then dichloromethane (18 mL). Sample vials were then washed with 500 µL dichloromethane an additional two times to ensure quantitative transfer and the combined 1 mL wash loaded onto the column by gravity. After washing the SPE bed with two column-volumes of dichloromethane (12 mL), samples were eluted with 18 mL 10% MeOH in dichloromethane, and the resultant fraction evaporated under a gentle stream of nitrogen at 40 °C. Prior to bioassay analysis, the SPE fractions of each sample were standardized to 10–184 g/mL tissue equivalents relative to the original fish tissue (Table 2) by dilution with MeOH. Aliquots from these were further diluted to 20 and 2 mg tissue equivalents (mgTE) and screened in the MTT-N2a assay as described elsewhere [17]. Since an MTT-N2a method comparison was not the focus of this study, samples were analyzed in a screening format (positive vs. non-detect) to simply verify CTX-like activity of extracts confirmed to contain C-CTXs by LC-HRMS. Characteristic CTX-like activity was evaluated in cells co-treated with ouabain and veratrine and considered positive if >30% loss in cell viability was observed compared to controls. Screening data below the 30% threshold at the stated dose was noted as a non-detect. Non-specific activity was evaluated in cells co-treated with PBS (no ouabain or veratrine) to monitor for potential matrix effects, but was not observed at the doses tested.

### 4.3. Derivatization of C-CTX-1 (***1***) in Fish Extracts

As pure reference materials were not available for method optimization, we selected five fish extracts (samples A–D and L) based on MTT–N2a screening. These were then shown to contain **1** and C-CTX-3/-4 (**2**) by LC–HRMS and comparison to authenticated extracts containing **1** and **2** that were available from previous work [17].

*Derivatization methods 1A and 1B.* To C-CTX-containing extracts (15 µL, equivalent to 0.15–2.76 g fish tissue) the following were added in the order listed: GRT (25 µL; 20 mg/mL in MeOH), 2PBC (15 µL; 10 mg/mL in MeOH), and 6 µL of TFA (Method 1A) or formic acid (Method 1B). Mixtures were left to react for 2 h at 60 °C.

*Derivatization method 2.* To C-CTX-containing extracts (15 µL; as above) the following reagents were added: GRT (7.5 µL; 1 mg/mL in MeOH), and formic acid in water (15 µL; 2.5% *v*/*v*). The mixture was shaken for 18 h at 20 °C, and thereafter sodium cyanoborohydride (7.5 µL; 2 mg/mL in MeOH) was added and allowed to react for 4 h at 40 °C.

*Derivatization method 3.* To C-CTX-containing extracts (15 µL; as above) the following were added in order: AETMA (25 µL; 10 mg/mL in water), 2PBC (15 µL; 10 mg/mL in MeOH), and acetic acid (6 µL). Mixtures were allowed to react for 2 h at 60 °C (Figure 2).

In general, reaction mixtures were analyzed by LC–HRMS method 1 immediately after derivatization, and without further purification. Derivatized samples were stored at −20 °C and were stable for several weeks based on re-analyses over time. For determination of MS signal enhancement, untreated fish extracts were diluted with MeOH to obtain the same concentration as in the reaction mixtures and analyzed by LC–HRMS method 1 alongside the derivatized aliquots. Control samples and reaction mixtures were stored at −20 °C and analyzed in three technical replicates.

### 4.4. LC–HRMS and LC–MS/MS

*LC–HRMS method 1.* This method was used for qualitative analyses of the products from the derivatization reaction, and to determine the signal enhancement. Analyses were performed on a Vanquish Horizon UHPLC instrument connected to a Q-Exactive Hybrid Quadrupole–Orbitrap mass spectrometer, equipped with a HESI-II heated electrospray interface (all Thermo Fisher Scientific, Waltham, MA, USA). Extracts containing C-CTXs and reaction mixtures (3 µL injections) were separated on a Kinetex F5 column (100 mm × 2.1 mm i.d., 1.7 µm; Phenomenex, Torrance, CA, USA). Analytes were eluted at 0.3 mL/min and 30 °C with a linear gradient of mobile phases A (0.1% formic acid in acetonitrile–water, 5:95, *v*/*v*) and B (0.1% formic acid in acetonitrile–water, 95:5, *v*/*v*), from 20% to 50% B over 14 min, followed by a column flush with 99% B for 3 min, and then returned to 20% B and equilibrated for 3 min. The mass spectrometer was set to full-scan in the mass range *m*/*z* 1050–1350 in positive ionization mode. Other important instrument parameters included mass resolution set to 140,000 (at 200 *m*/*z*), target ion count automatic gain control set to 1 × 10^6^, maximum ion inject time of 512 ms, an S-lens RF level of 100%, and an ESI voltage of 3.0 kV.

*LC–HRMS method 2.* This was identical to LC–HRMS method 1 except that a Kinetex EVO C18 column (100 mm × 2.1 mm i.d., 1.7 µm; Phenomenex) was used, and the gradient was adjusted to rise from 20% to 60% B over 14 min. This method was used to verify the performance of octadecylsilane (ODS) LC columns for the analysis of the derivatized extracts.

*LC–MS/MS method.* Analyses were performed on an Agilent 1290 Infinity II LC coupled to a 6495B mass spectrometer (Agilent Technologies, Missisauga, ON, Canada). Separation conditions were identical to LC–HRMS method 1. The mass spectrometer was set to MRM mode evaluating the transitions for **1** (*m*/*z* 1123.6 → 1105.6, CE 25 eV), **2** (*m*/*z* 1143.6 → 1107.6, CE 25 eV), and **4** (*m*/*z* 1256.7 → 60.4, CE 67 eV), with dwell times of 200 ms. Source conditions were as follows: gas temperature 220 °C, gas flow 11 L/min, nebulizer pressure 20 psi, sheath gas heater 400 °C, sheath gas flow 12 L/min, capillary 4000 V, nozzle voltage 0 V, high pressure RF 210, and low pressure RF 120.

### 4.5. Estimation of LC–HRMS and LC–MS/MS Signal Enhancement for ***1*** via Reductive Amination with GRT

The LC–HRMS signal enhancement for samples was calculated from extracted ion chromatograms (±5 ppm) as the ratio of the area of the base peak (PA) of the main epimer of C-CTX-1–GRT (**4**, [M]^+^, *m*/*z* 1256.7415) after derivatization and the peak area of the base peak of **1** (*m*/*z* [M+H–H_2_O]^+^, 1123.6200) in the corresponding untreated control sample (after dilution to make the concentration directly comparable to that of the derivatized sample):Signal enhancement = PA (**4**)/PA (**1**)

To further evaluate the sensitivity enhancement, and the degree to which detection of low levels of **1** in fish extracts can be improved by the derivatization procedure, dilutions of C-CTX-containing sample A were prepared in negative sample N (0%, 1%, 5%, 20%, 50%, and 100%). Derivatized (derivatization method 1A and method 1B) and non-derivatized aliquots of these mixtures were analyzed using LC–HRMS method 1.

In order to evaluate the performance of the reductive amination procedure for a tandem quadrupole mass spectrometer method, non-derivatized sample D was analyzed against the derivatized sample. The peak areas corresponding to the transitions for **1** and **4** were compared. Serial dilutions of the GRT-derivatization solution for sample D were prepared at 10-, 100-, 1000-, 10,000-, and 100,000-fold dilutions with MeOH to evaluate instrument response, sensitivity, and linearity. Instrument repeatability was assessed by injecting triplicate injections of GRT-derivatized sample D.

## Figures and Tables

**Figure 1 toxins-14-00399-f001:**
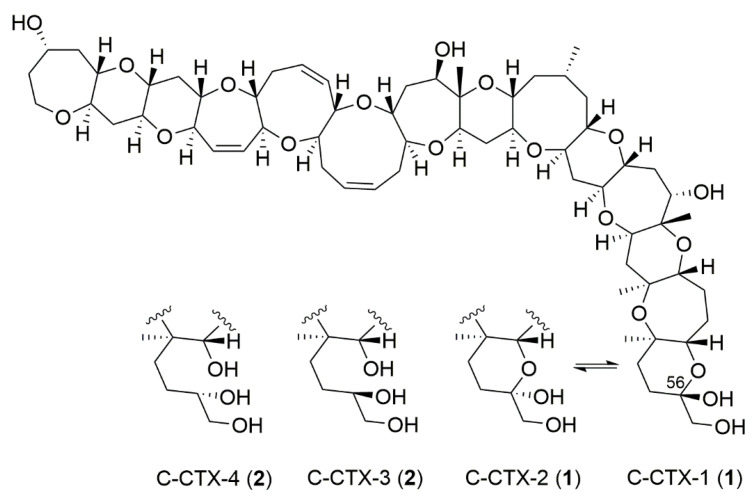
Chemical structures of the equilibrating pair of 56-epimers C-CTX-1 and C-CTX-2 (**1**), referred to as C-CTX-1 for simplicity, and its non-equilibrating reduced analogues, C-CTX-3 and C-CTX-4 (**2**). The structures of **1** are shown in accordance with Lewis et al. [18], and structures of **2** were reported previously [17].

**Figure 4 toxins-14-00399-f004:**
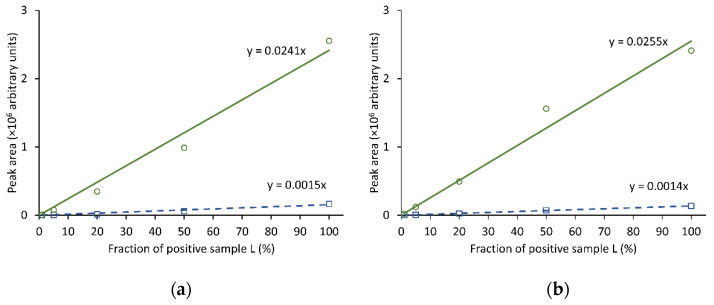
Instrument response achieved either by direct LC–HRMS analysis ((**a**), derivatization method 1A; (**b**), derivatization method 1B on the right) for **1** (*m*/*z* 1123.6200) of C-CTX-1-containing fish extracts (squares, blue dashed line) before, or for **4** (*m*/*z* 1256.7415) after, GRT derivatization (circles, green full line) using derivatization method 1. The data were obtained by mixing extracts of sample-L (0%, 1%, 5%, 20%, 50%, and 100%) with negative control fish extract-N, and show an increase of about 17-fold (**a**) (left) and 19-fold (**b**) in sensitivity after reductive amination with GRT.

**Figure 5 toxins-14-00399-f005:**
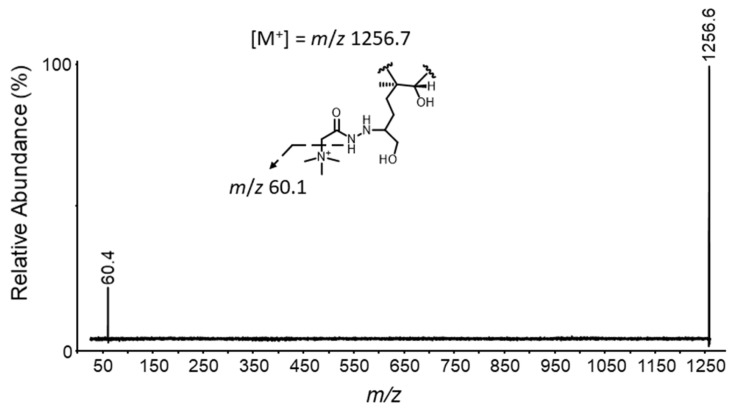
Triple quadrupole LC–MS/MS product ion spectrum of **4** (*m*/*z* 1256.7) with a collision energy of 67 eV in positive ionization mode.

**Figure 6 toxins-14-00399-f006:**
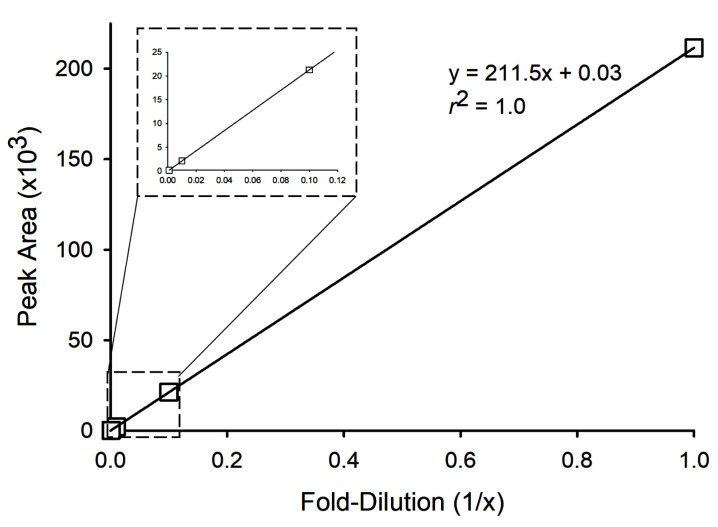
LC–MS/MS response for the serial dilutions of **4** in sample D demonstrating linearity across several orders of magnitude. The inset shows an expansion of the region for the highest dilutions.

**Table 1 toxins-14-00399-t001:** Molecular formulae, ring plus double-bond equivalents (RDBE), and retention times (RT) for LC–HRMS method 1, as well as observed *m*/*z* for main ions, and mass error (∆*_m_*) for compounds in this study.

Compound	Neutral Formula ^a^	RDBE	RT (min)	Ion	Accurate *m*/*z*	∆*_m_* (ppm)
C-CTX-1 (**1**)	C_62_H_92_O_19_	17	11.23	[M+H–H_2_O]^+^	1123.6232	+2.8
C-CTX-3/-4 (**2**)	C_62_H_94_O_19_	16	9.75	[M+H]^+^	1143.6493	+2.7
C-CTX-1–GRT (**3**)	C_67_H_104_N_3_O_19_^+^	18	8.84	M^+^	1254.7303	+3.6
Reduced C-CTX-1–GRT (**4**)	C_67_H_106_N_3_O_19_^+^	17	7.71	M^+^	1256.7449	+2.7
C-CTX-1 56-methyl ketal (**5**)	C_63_H_94_O_19_	17	13.07	[M+H–MeOH ^b^]^+^	1123.6231	+2.8
C-CTX-1 56-(2-methoxyethyl) ketal	C_65_H_98_O_20_	17	13.93	[M+H–C_3_H_8_O_2_]^+^	1123.6206	+0.5
C-CTX-1 56-(1-butyl) ketal	C_66_H_100_O_19_	17	15.34	[M+H–BuOH ^c^]^+^	1123.6208	+0.7

^a^ Except for **3** and **4**, which contain cationic trimethylammonium moieties (Figure 2). ^b^ Methanol. ^c^ 1-Butanol.

**Table 2 toxins-14-00399-t002:** Peak areas for C-CTX-1 (**1**) prior to derivatization, and of C-CTX-1-GRT (**4**) after reductive amination with GRT (*n* = 3) using derivatization method 1A and LC–HRMS method 1. Peak areas are from extracted ion chromatograms for **1** ([M+H–H_2_O]^+^), its GRT derivative **4** ([M]^+^), and **2** ([M+H]^+^), using LC–HRMS method 1.

	Tissue Equiv. (g/mL)	Peak Areas before Derivatization		Peak Areas after Derivatization		Signal Enhancement
		C-CTX-1 (1)	C-CTX-3/4 (2)	C-CTX-1-GRT (4)	C-CTX-3/4 (2)	
Sample A	40	2.58 × 10^5^	2.00 × 10^4^	3.48 × 10^6^	4.53 × 10^4^	13.5
Sample B	40	4.18 × 10^5^	7.48 × 10^4^	5.43 × 10^6^	1.01 × 10^5^	13.0
Sample C	40	5.06 × 10^5^	8.06 × 10^4^	7.37 × 10^6^	1.13 × 10^5^	14.6
Sample D	40	2.71 × 10^6^	4.60 × 10^5^	4.95 × 10^7^	6.81 × 10^5^	18.2
Sample E	24	6.12 × 10^3^	(1.32 × 10^3^) ^a^	1.38 × 10^5^	(1.62 × 10^3^) ^a^	22.6
Sample F	24	1.59 × 10^4^	(2.64 × 10^3^) ^a^	2.88 × 10^5^	(4.84 × 10^3^) ^a^	18.1
Sample G	24	2.35 × 10^4^	(2.89 × 10^3^) ^a^	2.82 × 10^5^	(4.18 × 10^3^) ^a^	12.0
Sample H	168	(2.25 × 10^3^) ^a^	(1.61 × 10^3^) ^a^	2.43 × 10^4^	(1.45 × 10^3^) ^a^	10.8
Sample I	184	(4.73 × 10^2^) ^a^	-	7.97 × 10^3^	-	16.8
Sample J	25.2	(4.82 × 10^3^) ^a^	-	4.51 × 10^4^	-	9.4
Sample K	24.2	9.32 × 10^3^	-	7.08 × 10^4^	-	7.6
Sample L	10	2.43 × 10^5^	2.87 × 10^4^	3.82 × 10^6^	4.69 × 10^4^	15.7
Sample N	10	0	0	0	0	-

^a^ Peak areas in parentheses were below the LOQ, which was estimated to be 5 × 10^3^ for a signal-to-noise ratio (S/N) of 10.

**Table 3 toxins-14-00399-t003:** Sample ID and origin of fish prepared and analyzed in this study. Note that different sample IDs for the same species from a given region refer to independent fish.

Sample ID	Genus. Species ^a^	Common Name	Region of Collection	Extraction Method	MTT-N2a Screening ^c^	
					20 mg TE	2 mg TE
A	*S. cavalla*	King Mackerel	St. Thomas, USVI	1	(+)	(−)
B	*S. regalis*	Cero Mackerel	St. Thomas, USVI	1	(+)	(+)
C	*S. barracuda*	Great barracuda	St. Thomas, USVI	1	(+)	(+)
D	*S. barracuda*	Great barracuda	St. Thomas, USVI	1	(+)	(+)
E	*S. cavalla*	King Mackerel	Puerto Rico	2	(+)	(−)
F	*S. cavalla*	King Mackerel	Puerto Rico	2	(+)	(−)
G	*S. cavalla*	King Mackerel	Puerto Rico	2	(+)	(−)
H	*S. barracuda*	Great barracuda	St. Thomas, USVI	2	(+)	(+)
I	*S. barracuda*	Great barracuda	St. Thomas, USVI	2	(+)	(+)
J	*S. barracuda*	Great barracuda	St. Thomas, USVI VI	2	(+)	(+)
K	*S. cavalla*	King Mackerel	St. Thomas, USVI VI	2	(+)	(−)
L	*S. barracuda*	Great barracuda	St. Thomas, USVI Virgin Islands	1	(+)	(+)
N ^b^	*S. barracuda*	Great barracuda	Dauphin Island, US	3	(−)	(−)

^a^ Species identification by morphometric and barcoding methods. ^b^ Negative sample. ^c^ MTT-N2A performed in a screening format at 20 mg and 2 mg tissue equivalent (mgTE) doses, performed as previously reported [17,32]. Non-specific activity was not detected in any sample, and thus ouabain–veratrine-sensitized response is reported. Positive CTX-like response exceeding 30% reduction in cell viability compared to controls is indicated by a (+) and responses below this threshold are indicated with (−).

## Data Availability

All necessary data are shown either in the manuscript or the Appendix A.

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
