# Peer review of "Reductive Amination for LC–MS Signal Enhancement and Confirmation of the Presence of Caribbean Ciguatoxin-1 in Fish"

_toxins, 2022, doi:10.3390/toxins14060399_

Round 1

Reviewer 1 Report

Review of manuscript “toxins-1742505-peer-review-v1.pdf”, Reductive amination for LC–MS signal enhancement and confirmation of the presence of Caribbean ciguatoxin-1 

This well written manuscript points out the need for developing more sensitive methods for detecting regulatory levels of ciguatoxins in fish and shellfish. The authors for the first time introduce the use of Girard’s reagent T (GRT) to tag the C-56-ketone intermediate of the two equilibrating C-56 epimers of C-CTX-1 with a quaternary ammonium moiety. This tagging significantly increases the detection sensitivity of C-CTXs using LC–MS/MS or LC–HRMS. The study shows this increased sensitivity makes it possible to detect C-CTX-1,2 (the primary ciguatoxins found in Atlantic fishes) at levels below the regulatory limit. This represents a major advance in the field. 

That said, the abstract and conclusions make the results sound more definitive than they are in actuality. Table 2 shows there is a 3-fold variation in signal enhancement. This is likely due to differences in the conversion efficiency. It is suggested the authors make it clearer that this significant advance requires further refinement with respect to controlling the conversion reaction between GRT and C-CTX-1 before the new method can be employed for routine detection of ciguatoxins.

The authors also need to describe how the fish samples were screened for potential CTX content using N2a assay in greater detail and to express the estimated CTX content of the fish in CTX3C or some other equivalents. The Table 3 legend indicates these data exist and if so, the data for each fish should be included in either Table 3 or a new figure showing the correlation between the N2a assay and the derived GRT peak areas. Without these data there is no way the current study can be interpreted with respect to previous N2a or LC-MS studies. There is also no quantitative estimate of recovery efficiency provided for the various isolation methods. These data are critical. In conclusion, publication of the manuscript is recommended, but only after revision to address the issues raised above. 

Other minor suggested edits include the following:

Lines

19-22   Modify the abstract to indicate this is an important advance with regard to detecting Caribbean ciguatoxins, but that further work is needed to optimize the conversion efficiency. 

32        Suggest “health impacts to temperate regions through export of contaminated fish.”

57-67  The description of the N2a assays  is confusing. Suggest the following outline.

            N2A is a cytotoxicity assay. 

            For 24 h cell layer grown to near confluence on well plate

            N2A cells are relatively insensitive to CTX toxicity

            Hence add veratridine which binds site 2 on the voltage-gated sodium channel, sensitizing the channels to open more easily. Ouabain is added to inhibiting the Na+/K+–ATPase sodium-potassium ion pump which pumps pumping ions out of the cell  

            CTXs bind site 5 on the already sensitized sodium channels causing an influx of sodium into the cell setting off an apoptotic cascade. The resulting amount of cell death is proportional to the amount and type of CTXs present in the sample

            After a 24 h incubation MTT, a mitochondrial dye whose indicates whether cell are alive or dead is added and read using a spectrophotometer. 

            Standard curves are produced using isolated CTXs and is highly sensitive to CTXs at levels below regulatory limits

            The assay is most commonly used to detect CTXs but can be used to detect other neurotoxins that target sodium channels such as brevetoxins. 

            Note: recent work has shown maitotoxin 4 will trigger N2A assay indicating the assay is not as specific as previously believed. 

Get the information from a previous manuscript

Fig. 1   The structural difference between C-CTX-3 ad C-CTX-4 is not clear.  Add distinct figures for each structure to keep the information consistent with the diagram shown for C-CTX-1 and C-CTX-2.

83 and 320.   ug/kg or ug kg-1?

118 –   Implication of the exception -    * Except for 3 and 4, which contain trimethylammonium moieties is not clear.

210      State what concentrations of C-CTX. You were starting with.  What constitutes the concentrations observed?

296, 297 italicize species names

352-354     How the recovery efficiency was determined should be described in more detail and is not beyond the scope of the current study. 

384      mg/ml or mg mL-1?

404      ml/min or mL min-1?

423      L/min or min-1?

Figure 4. Colors of the dashed line were indistinguishable in my copy of the manuscript. Suggest solid versus dashed regression lines.

Table 2 Why is the signal enhancement ratio so variable (3-fold) – differences in the conversion efficiency?

Reviewer 2 Report

A brief summary

Potent toxic substances ciguatoxins (CTXs), produced by unicellular marine dinoflagellates belonging to the Gambierdiscus genus, can cause poisoning to a person who consumes fish or shellfish contaminated with these toxins. CTXs are difficult to detect due to their low content even in highly toxic fish. The authors of this work have proposed a method for increasing the sensitivity of the detection of CTXs using a fast and simple one-pot derivatization with Girard’s reagent T (GRT) that tags the C-56-ketone intermediate of the two equilibrating C-56 epimers of Caribbean-CTX-1 with a quaternary ammonium moiety. This derivatization improved the LC–MS/MS and LC–HRMS responses to C-CTX-1 by approximately 40- and 17-fold, respectively.

Broad comments 

The reaction scheme, optimization of the reductive amination of C-CTX-1, signal enhancement of C-CTX-1 using GRT, sample preparation methods, and measurement procedures are presented. This strategy to increase the sensitivity of the detection of CTXs can easily be applied to other regions (Indian and Pacific).

Specific comments 

Line 120               ODS-based         reveal abbreviation

Line 160               TFA        è          Trifluoroacetic acid (TFA)

Line 177               2PBC     reveal abbreviation

Author Response

Reviewer#2

A brief summary

Potent toxic substances ciguatoxins (CTXs), produced by unicellular marine dinoflagellates belonging to the Gambierdiscus genus, can cause poisoning to a person who consumes fish or shellfish contaminated with these toxins. CTXs are difficult to detect due to their low content even in highly toxic fish. The authors of this work have proposed a method for increasing the sensitivity of the detection of CTXs using a fast and simple one-pot derivatization with Girard’s reagent T (GRT) that tags the C-56-ketone intermediate of the two equilibrating C-56 epimers of Caribbean-CTX-1 with a quaternary ammonium moiety. This derivatization improved the LC–MS/MS and LC–HRMS responses to C-CTX-1 by approximately 40- and 17-fold, respectively.

Broad comments 

The reaction scheme, optimization of the reductive amination of C-CTX-1, signal enhancement of C-CTX-1 using GRT, sample preparation methods, and measurement procedures are presented. This strategy to increase the sensitivity of the detection of CTXs can easily be applied to other regions (Indian and Pacific).

Specific comments 

Line 120              ODS-based         reveal abbreviation

Abbreviation explained at first mentioning.

Line 160              TFA        è          Trifluoroacetic acid (TFA)

Abbreviation explained at first mentioning.

Line 177              2PBC     reveal abbreviation

Abbreviation explained at first mentioning.

Reviewer 3 Report

I am hesitant to choose a rating for the 'overall merit' of the manuscript. While the presentation and discussions of the results is very interesting and well done, the methods are not very clear and must be improved.

After I read the article, I concluded that it is well written, almost clear, comprehensive and relatively easy to understand.

The article start with an introduction in the field of algal-derived ciguatoxins (CTXs) and in vitro bioassays for CTX detection.

The article continue with the obtained results and some discussion showing that for the improvement of ionization efficiency, one way of work consists in derivatization of CTXs with different reagents (2-aminoethyl)tri-292 methylammonium chloride hydrochloride - AETMA and  Girard’s reagent T – GRT). The sections regarding the optimization of the reductive amination of C-CTX-1 and C-CTX-2, and the signal enhancement using GRT were very interesting, well done. I want to congratulate the authors for these two sections.

The conclusions were supported by the results.

The last section contains the used materials and methods.

The statements and conclusions are coherent and supported by the listed citations. The figures presented are appropriate; they properly show the data and are easy to interpret and understand.

Level of interest: An article with importance in its field but not only, because it offers a variant of increasing the detection sensitivity that can be applied in other types of determinations or research fields.

Conclusions

I am hesitant to choose a rating for the 'overall merit' of the manuscript. While the presentation and discussions of the results is very interesting and well done, the methods are not very clear and must be improved.

As final conclusions, after studying the article and reading for many times, I found in all the article 28 minor problems for revision. But, in Material and Methods section, I found also 8 problems for which some completions and explanations are necessary and I consider to be major.

 Provided that the issues listed as minor and major revisions are settled, the paper is worthy of publication.

Below are my comments and the main points that I think should be reviewed.

Minor revisions

1)             Lines 31, 50, 100, 122 – “Figure 1”: make a link to Figure 1 at line 52

2)             Lines 98, 100-101, 214 – “Figure 3V”, “Figure 3II” and “Figure 3”: make a link to Figure 3 at line 225

3)             Lines 124, 128, 144, 158, 171, 182, 389 – “Figure 2”: make a link to Figure 2 at line 113

4)             Lines 145, 215 – “Figure S2”; Lines 152 – “Figure S1”; Lines 157, 158, 168-169 – “Figure S4”; Lines 158, 160, 164 – “Figures S5”; Line 187 – “Figure S6”; Lines 193-194 – “Figure S7”; Line 201 –  “Figure S3”; Lines 254 – “Figure S9”: Because these figures are shown in the supplementary materials, please specify this

5)             Lines 217, 221 – “Figure 4”: make a link to Figure 4 at line 231

6)             Line 247 – “Figure 5”: make a link to Figure 5 at line 242

7)             Line 260 – “Figure 6”: make a link to Figure 6 at line 266

8)             Line 128 – “Table 1”: make a link to Table 1 at Line 116

9)             Lines 137, 173 – “Table S1A–S1D”: Write “Tables” instead of “Table”. Because these tables are shown in the supplementary materials, please specify this.

10)          Lines 206, 223, 323, 372 – “Table 2”: make a link to Table 2 at line 236

11)          Lines 299, 316 – “Table 3”: make a link to Table 3 at line 318

12)          Line 52, “C-CTX”: Here, this abbreviation appear for the first time in figure legend. Explain what means (Caribbean CTXs).

13)          Line 66, “N2a-MTT”: Maintain the same format for abbreviation as before “MTT-N2a” at line 60 where was defined first time.

14)          Line 82: Write LOQ instead of LoQ

15)          Line 115 and 123, “AETMA”: Here, this abbreviation appear for the first time (in figure legend – line 115 and in text – line 123). Explain what means ((2-aminoethyl)tri-292 methylammonium chloride hydrochloride).

16)          Line 118, in table 1, “MeOH”, and “BuOH”: Here, this abbreviation appear for the first time. Explain what means (methanol and butanol respectively).

17)          Line 160, “TFA”: Here, this abbreviation appear for the first time. Explain what means (Trifluoroacetic acid).

18)          Line 177, “2PBC”: Here, this abbreviation appear for the first time. Explain what means (2-methylpyridine borane complex).

19)          Lines 236-239, Table 2: Here is a ambiguity: in table 2 header are presented the peak areas for C-CTX-3/4 (2) before derivatization and C-CTX-3/4 (2) after derivatization. Maybe I make a mistake, but shouldn’t appear another notation for C-CTX-3/4 (2) after derivatization? At first glance the same notation creates an ambiguity.

20)          Lines 260-261, “sample (Figure 6) and repeatability of replicate injections had a relative standard deviation of 0.5% in sample L.”: for how many replicate injections?

21)          Lines 297-298: Write “Scomberomorus cavalla”, “Scomberomorus regalis” and “Sphyraena barracuda” with italic

22)          Line 324: I suggest “5 mL/g tissue” instead of “5 mL/g”.

23)          Lines 328-329 – “The residual tissue recovered from the filters was then extracted an additional two times”: I suppose with the same solvent (acetone). Please specify the solvent and the used volume.

24) Line 334, partitioned with chloroform (3 × 0.5 vol.): I’m a little bit confused. What means 3 x 0.5 vol.? I suppose that the 50% methanolic phase was partitioned 3 times with chloroform. But 0.5 vol.? I suggest rephrase for a better understanding. Please specify.

25) Line 346: The same as before. What means 2 x 0.6 vol.? Please specify.

26) Line 348: The same as before. What means 3 x 0.5 vol.? Please specify.

27)          Line 366, “Sample vials were then further washed with two 500 μL aliquots of dichloromethane 366 and loaded onto the column by gravity”: It's a bit ambiguous, I suggest "Sample vials were then further washed twice with 500 μL dichloromethane and the combined washing solutions are and loaded onto the column by gravity".

28)          Line 387, “Derivatization method 3. To ciguatoxic extracts (15 μL)”: Write “(15 μL; as above)”.

Major revisions (4. Materials and Methods)

1)             Lines 318-322,Table 3: For this study were used samples: A – L, from which five (A – D, L) were extracted by procedure 1 and seven (E–K) by procedure 2.

a)     Which is the difference between samples from the same genus and species, same location and same extraction procedure (C–D, E–G, H–J)? Please specify. (For example, samples E, F and G respectively are the same genus and species (S. cavalla), from same location (Puerto Rico) and processed by the same extraction procedure (procedure 2). Which is the difference between them? I ask this because, for these samples, in table 2 it is shown that for the same tissue equivalent (24 g/mL) were obtained different peak areas both before and after derivatization). Please specify.

b)    From samples extracted by procedure 2 (E–K) there are some that are extracted by procedure 1? If so, please specify the correspondence. If not, also specify.

c)     Samples B and L are the same genus and species (S. regalis), same location (St. Thomas, US Virgin Islands) and the extraction procedure is the same (1). What to understand:

-       there is a mistake – extraction procedure 2 instead of 1 for sample L?

-       if the samples and the extraction procedure are the same, why in table 2 for a tissue equivalent of 40 g/mL for sample B and 10 g/mL for sample L, the results in peak areas (for example C-CTX-1) before (4.18x105 for B and 2.43x105 for L) and after derivatization (5.43x106 for B and 3.82x106 for L) are not proportional as we can believe from figure 4 where the instrument response (peak area) seems to be relatively linear, as well as from figure 6?

-       or these two samples are different. In this case, which is the difference, in addition to the tissue equivalent in g/mL (question from paragraph a)?

2)             Line 323, “Samples were extracted using one of three methods (Table 2)”: Two aspects:

a)    Table 2 summarizes details of the signal enhancement, including peak areas prior to derivatization (from direct analysis), and after derivatization (Lines 206-208 and 236-239) and does not refer to extraction methods. I think that here is good to send a link to Table 3 not for Table 2.

b)    Reading lines 220 – 223: “The signal enhancements observed in the dilution study (Figure 4) suggest that derivatization of 1 with GRT resulted in an approximately 17-fold increase in analyte peak area, consistent with the enhancements observed for the five most contaminated extracts (samples A–D, and L) in Table 2.”, I understand that the extraction method 1 is more sensitive that the extraction method 2. In this case, taking into account the fact that the ultimate goal is to increase the detection sensitivity, why it was used the second extraction method. Or, another possibility: taking into account the fact that different solvents are used in the two liquid phase extraction procedures, it is possible that procedure 1 has a higher extraction efficiency and as a result for the samples processed by this method to obtain larger peak areas so as to they seem to be more contaminated. As a conclusion, I think that more explanations are needed.

3)             Lines 339-346: How we can know how much water must added to get a concentration of 80% in methanol if the extract contains approximately 15% water? Has the methanol content been determined?

4)             Line 346: The same as before. How we can know how much water must added to get a concentration of 50% in methanol?

5)             Lines 323-362: The extraction procedures were used for different samples (A, B, C, D, L – Extraction 1; E, F, G, H, K, I, J, K – extraction 2; N –extraction 3 as is shown in Table 3), but the procedures are not completely described or are not very clear. See also Comments 9 and 10.

a)     In the first procedure the tissue is extracted in acetone (three times), the combined filtrates is evaporated and dissolved in methanol and, after cleaning with hexane, is extracted in chloroform which was evaporated. At the end it remains a residue. What happens with the dried residues?

b)    In the second procedure the tissue is extracted in methanol (two times), the filtrate is adjusted with water and, after cleaning with hexane, is adjusted again with water and extracted in dichloromethane (three times) and the combined dichloromethane phases were evaporated. So far, the procedure is similar to the first. The residue is then re-dissolved in dichloromethane and evaporated again. What happens next with the last residue?

c)     In the third procedure the tissue is extracted in a mixture of acetone/water, centrifuged and evaporated. The residue is dissolved in methanol, cleaned with hexane, and after evaporation the residue is dissolved in 2 mL dichloromethane. At the end it remains a solution. Why a solution and not a residue, like in the procedures 1 and 2?

6)             Line 363, “SPE fractionation: The dichloromethane solutions were loaded”: For SPE fractionation were used solutions in dichloromethane, but in extraction procedures 1 and 2 were obtained only some residues. This is the reason for which I think that the extraction procedures 1 and 2 are not completely described. See also comment 10.

7)             Line 366, Sample vials were then further washed with two 500 μL aliquots of dichloromethane: Two aspects

a)      If the residue obtained by extraction procedure 1 or 2 is washed with two 500 μL aliquots of dichloromethane, as is presented at line 366, this must be specified at the end of the extraction procedure, something like “the final residue will be recovered with dichloromethane for SPE fractionation” or “those residues are ready for SPE fractionation“ or something like this.

b)    But another question appear: in the third procedure it was obtained 2 mL solution in dichloromethane, not a residue. How to wash 2 mL solution with 500 μL? I suggest to rephrase the paragraphs for extraction procedures and SPE fractionation, in order to be very clear how the experiments were conducted.

8)             Line 445, “Instrument repeatability was assessed by injecting triplicate injections of GRT-derivatized sample L.”: According ICH Topic Q 2 (R1) Validation of Analytical Procedures: Text and Methodology, Note for guidance on validation of analytical procedures: text and methodology (CPMP/ICH/381/95), June 1995 and ICH guideline Q2(R2) on validation of analytical procedures, Step 2b, 31 March 2022 EMA/CHMP/ICH/82072/2006, repeatability should be assessed using: a) a minimum of 9 determinations covering the specified range for the procedure (e.g. 3 concentrations/ 3 replicates each) or a) a minimum of 6 determinations at 100% of the test concentration. So, triplicate injections are not enough.

Round 2

Reviewer 3 Report

 After I read the authors' responses and the revised article, I came to the conclusion that the authors made the suggested changes or extended the necessary explanations.